# A new external jugular venipuncture technique for efficient vascular access that exploits a murine anatomical variation

Suguru Yamauchi[1,2,3], Andrei Gurau[1], Kaitlyn Ecoff[1], Kristen P. Rodgers[1], Yuping Mei[1], Frank Bosmans[4,5], Franck Housseau[6], Yun Chen[7], John Michel[6], Andreas S. Barth[8], Jinny S. Ha[9], Takumi Iwasawa[10], Kazunori Kato[10], Ryohma Tsuchiya[7], Miki Yamauchi[3], Hajime Orita[2,3], Shinji Mine[2], Tetsu Fukunaga[2], Malcolm V. Brock[1]*

1 Department of Surgery, Johns Hopkins University School of Medicine, Baltimore, Maryland, United States of America, 2 Department of Esophageal and Gastroenterological Surgery, Faculty of Medicine, Juntendo University, Tokyo, Japan, 3 International Collaborative Research Administration, Juntendo University School of Medicine, Tokyo, Japan, 4 Molecular Physiology and Neurophysics group, Department of Basic and Applied Medical Sciences, Faculty of Medicine and Public Health, University of Ghent, Ghent, Belgium, 5 Experimental Pharmacology group (EFAR), Department of Pharmaceutical Sciences, Faculty of Medicine and Pharmaceutical Sciences, Vrije Universiteit Brussel, Brussel, Belgium, 6 Department of Oncology and Sidney Kimmel Comprehensive Cancer Center, Johns Hopkins University School of Medicine, Baltimore, Maryland, United States of America, 7 Department of Mechanical Engineering, Johns Hopkins University, Baltimore, Maryland, United States of America, 8 Department of Medicine, Division of Cardiology, Johns Hopkins University School of Medicine, Baltimore, Maryland, United States of America, 9 Division of Thoracic Surgery, Johns Hopkins University School of Medicine, Baltimore, Maryland, United States of America, 10 Institute of Life Innovation Studies, Toyo University, Tokyo, Japan,

* mabrock@jhmi.edu

## Abstract

Vascular access in mice is a cornerstone of biomedical research, with peripheral venous approaches like the lateral tail vein, retrobulbar venous sinus, facial vein, and saphenous vein being common. However, central venous approaches are challenging due to animal size and required expertise. To address this, we developed the Sternoclavicular joint-Targeted External jugular venipuncture Method (STEM). This technique provides reliable, longitudinal vascular access for frequent blood sampling using palpable surface anatomy landmarks. Moreover, STEM eliminates the need for fur shaving, specialized restraints, or deep sedation, allowing a single operator to perform the procedure safely and efficiently. Our protocol, based on a comprehensive anatomical analysis, revealed that the external jugular vein in mice traverses anteriorly to the clavicle before draining into the subclavian vein – a key anatomical difference from humans. This finding enabled a refined technique using the sternoclavicular joint as a landmark, improving the success and reproducibility of central venous access. Finally, STEM facilitates efficient blood collection and accurate intravenous administration with minimal setup time. It is straightforward and easily replicable, allowing researchers of all expertise levels to achieve high precision and reproducibility. The simplified learning process and consistent results make STEM valuable for various mouse-based experiments in biomedical research.

**Data availability statement:** All relevant data are within the paper and its Supporting information files.

**Funding:** SY is funded in part by the Subsidies for Current Expenditures to Private Institutions of Higher Education from the Promotion and Mutual Aid Corporation for Private Schools of Japan. This work was partly funded by a research grant from Dysautonomia International (90097658) to MB and FB. The work was also partly funded by the NIH under R01NS126398 (MB and FB), the Banks Family Foundation, Bermuda (MB) and the Skalka-Kronsberg family (MB). AG is funded by the NIH T32CA126607 Award. JM is funded by the NIH Medical Scientist Training Program grant T32GM136577. The funders had no role in study design, data collection and analysis, decision to publish, or preparation of the manuscript.

**Competing interests:** The authors have declared that no competing interests exist.

## Introduction

Mice are a preferred preclinical model for studying human biology because they share 85% of human genes and have similar anatomy, physiology, and pathophysiology [1]. Additionally, their ease of handling, rapid breeding, and short lifespan make them ideal for research. However, their small size makes chronic intravenous (IV) injections challenging due to the difficulty of locating and accessing their blood vessels. This is especially true if regular blood vessel access is required or if longitudinal studies necessitate frequent blood sampling. Mechanical restraints are often necessary, and proficiency in blood access requires considerable technical skill and experience.

Blood collection from mice often requires consideration of sample volume and frequency to avoid physiological stress or death. Longitudinal studies have used peripheral veins such as the lateral tail vein [2,3], retrobulbar venous sinus [4–6], facial vein [5,7,8], and saphenous vein [9,10]. However, these methods are limited by the fragility of peripheral veins, time consumption, and low blood perfusion of the murine peripheral veins.

The lateral tail vein is the most used method for accessing blood vessels in mice, with successful cannulation strongly depending on operator skill and experience [11]. Visualization and cannulation of the lateral tail vein are particularly challenging in juvenile mice and pigmented strains. Techniques like thermal vasodilation, warm water immersion, and topical alcohol application are often recommended to facilitate vein access, but their efficacy may be limited and pose risks of heat-related distress and injuries [12]. Other access sites, such as the retrobulbar venous sinus, allow for larger blood volumes but carry significant risks of ocular tissue damage [13–15]. Repeated sampling from the retro-orbital sinus can result in adverse effects, including intraocular hemorrhage, corneal ulceration, globe rupture, and optic nerve injury, potentially causing visual impairment or blindness [16]. There is also a lack of consensus on the degree of pain and the extent of histopathological damage to periocular tissues induced by retro-orbital puncture. Facial (submandibular) venipuncture has emerged as an alternative, particularly where retro-orbital bleeding is restricted to terminal procedures due to animal welfare concerns [5]. This technique, involving perforating the facial vein near the mandible, allows rapid blood collection without anesthesia but carries potential risks of inner ear or facial nerve/muscle damage if not performed precisely [17]. Consequently, venous blood collection from either the retrobulbar venous sinus or facial vein presents significant technical challenges for inexperienced operators and risks procedural complications and physiological stress responses that can negatively impact animal welfare. The saphenous venipuncture technique involves puncturing the lateral saphenous vein at the tarsus with a small-gauge needle to collect blood samples from conscious mice. This method allows rapid, serial blood collection but requires fur removal at the venipuncture site. While the superficial location of the saphenous vein enables precise access and visual confirmation of hemostasis, visualization can be challenging in juvenile mice or pigmented strains, sometimes necessitating blind venipuncture [10]. The main complication is persistent, low-volume hemorrhage, though the technique is generally considered safe [13]. Other limitations

include the need for physical restraint, which can induce significant stress, and the often lower-than-anticipated blood volume, occasionally requiring multiple venipunctures to achieve the required sample volume.

In contrast, central venous access, which allows for larger blood volumes with larger vascular diameter and higher blood flow has received less attention. Despite potentially beneficial venipuncture methods, central venous catheterization via jugular and subclavian veins requires deep sedation, trunk restraint, and fur shaving, which are cumbersome and time-consuming nature [18–20]. Jugular venous access has been described in a few reports [21–26] whereas subclavian venous access has been rarely reported [27]. The lack of detailed, step-by-step methods for central venous access from either the jugular or subclavian veins, and the lack of safety and efficacy of these techniques, has limited their adoption.

Experiments requiring IV injections in mice commonly involve venous cannulation of the lateral tail vein [11,28] and other peripheral veins [10,28–32], or catheter placement in the external jugular vein (EJV) [33–35]. However, multiple IV bolus doses via the lateral tail vein present technical challenges, including variability in the depth of the vein and difficulty in visualizing the needle entry point, particularly in mice with dark skin pigmentation. Additionally, there is no reliable method for confirming successful intravascular delivery versus inadvertent perivascular administration. These methods require advanced surgical skills, are prone to complications, and may not consistently provide accurate results. Procedural difficulties may also limit experimental design, potentially impacting the validity of any experimental results.

Despite the long-documented history of mouse experimentation, standardized methods for accessing murine blood vessels for blood sampling and IV injections remain a challenge. To address this issue, we developed a new venipuncture protocol focused on the EJV using an easily identifiable and palpable landmark – the sternoclavicular joint. We coined this method the "Sternoclavicular joint-Targeted External jugular venipuncture Method (STEM)". STEM provides a routine, standardized, and accessible route to a central murine vein, allowing for EJV venipuncture with a substantial margin of error, and high procedural success rates. The method is efficient and safe, enabling large blood volume collection in a highly reproducible manner with minimal sedation. It also can be performed single-handedly without specialized restraint equipment or fur removal. STEM is equally advantageous for IV fluid administration. It provides clear visual confirmation that injected substances are entering the bloodstream, which makes it possible to perform multiple injections on the same mouse. The EJV also offers a distinct advantage as a central venous access route due to its large caliber, accommodating concentrated injections and providing alternative access sites for serial injections.

STEM relies on newly discovered anatomical details from mouse autopsies. By using the sternoclavicular joint as a guide, this method improves the accuracy and consistency of needle placement for blood vessel punctures in mice. This innovative method has the potential to enhance significantly the use of daily blood access procedures in mice, particularly for experiments requiring repetitive, longitudinal blood sampling or IV injections. Here, we provide a detailed protocol accompanied by supplemental educational materials, including high-resolution narrated video demonstrations, designed to facilitate the seamless adoption of this technique for both novice researchers as well as experienced practitioners.

## Materials and methods

The protocol described in this peer-reviewed article is published on protocols.io (https://dx.doi.org/10.17504/protocols.io.eq2lywqkevx9/v1) and is included for printing as S1 File. In addition, our published protocol.io provides high-resolution narrated videos on STEM with separate tutorials for accessing the right and left EJV veins as well as demonstrations of IV access via the EJV. These resources are designed to facilitate understanding and mastery of all steps of this new technique. The comprehensive imaging methodology implemented in this study and the experiments performed to establish the technique are described in detail in the S2 Text.

### Mice

All experiments in this study were approved by the Institutional Animal Care and Use Committee of the Johns Hopkins University (MO23M72) and were conducted in compliance both with the Animal Research: Reporting of *In Vivo*

Experiments (ARRIVE) guidelines version 2.0 [36] and with the National Institutes of Health (NIH) Guide for the Care and Use of Laboratory Animals [37]. All studies made the greatest effort to minimize the suffering of mice. Experiments were performed using adult C57BL/6J, BALB/cJ, NOD/ShiLtJ and DBA2J mice of both sexes (8–12 weeks old). Mice were housed under a reversed 12:12 hour, light: dark cycle and provided with food and water ad libitum.

## Statistical analysis

Statistical analyses were performed using IBM SPSS Statistics for Mac, version 29.0.2.0 (IBM Corp., Armonk, N.Y., USA). The Shapiro-Wilk test was used to assess normality of continuous variables. Normally distributed variables were reported as mean ± standard deviation, while non-normally distributed variables were reported as median [interquartile range (IQR)]. The Levene's test was used to assess the homogeneity of variances. Comparisons between two groups were analyzed with the student t-test. For comparisons among three groups, one-way ANOVA or Kruskal-Wallis test was employed based on data distribution. When significant differences were found, appropriate post-hoc tests were conducted (Tukey's HSD for ANOVA, Dunn's test with Bonferroni correction for Kruskal-Wallis). The Chi-square test was applied for categorical variables. A p-value of <0.05 was considered statistically significant.

## Expected results and discussion

We have developed a new method, STEM, that addresses the challenges of repetitive, longitudinal blood collection and IV injection in mice. STEM eliminates the need for fur removal and specialized restraint equipment, allowing for single-hand immobilization and identification of anatomical landmarks without shaving. This technique enables rapid, large-volume blood collection via venipuncture which can be performed safely with minimal training. STEM is applicable to both blood sampling as well as IV access. In addition, this approach reduces the need for copious supplies and long setup times compared to traditional methods, thus streamlining the procedure and minimizing stress for the animals.

Traditional approaches for jugular vein blood collection in mice involve advancing the needle superior to the sternoclavicular joint in a caudal-to-cephalic direction [21,22,25,26]. This method presents challenges, especially in pigmented strains where visualizing the jugular veins can be difficult due to individual anatomical variations. Additionally, the lack of a standardized puncture site and variable EJV anatomy in mice necessitate considerable expertise to determine the optimal needle entry angle and direction. We hypothesized that the previous low success rate and poor reproducibility of central vein blood draws and IV injections were due to insufficient anatomical knowledge. To address this gap, we conducted an in-depth anatomical analysis to validate external jugular venipuncture using the sternoclavicular joint as a landmark. We chose the sternoclavicular joint for its consistent anatomical relationship with the EJV.

Our detailed imaging studies, including high-resolution ultrasound (Fig 1), contrast CT scanning (Fig 2), multi-planar reconstruction images (Fig 3), 3D volume rendering (Fig 4), and autopsy findings (Figs 5 and 6), revealed a significant anatomical variation of the murine EJV compared to humans. The EJV transverses anteriorly to the clavicle from the neck before draining into the subclavian vein and superior vena cava (Figs 2–5). Concomitantly, the EJV increases in diameter along its course, reaching about 2 mm near the clavicle (Figs 1–4). These key anatomical findings regarding the trajectory of the EJV in mice were consistent not only in the C57BL/6J strain but also in other mouse strains, including BALB/cJ, NOD/ShiLtJ, and DBA/2J (Fig 6).

Palpation and visual inspection of the mice also revealed a consistent triangular depression formed by the sternum and clavicle, aligning with optimal puncture sites identified through 3D CT reconstructions and cadaveric dissections (Figs 4–6). Real-time imaging shows needle advancement along the clavicular axis consistently intersecting the EJV at its maximal caliber after penetrating the fascial layers of the pectoralis major before entering the EJV lumen (Figs 5 and 6). Our venipuncture method leverages hemostatic compression from the pectoralis major, a significant advantage over traditional neck puncture of the non-adherent EJV. High-resolution imaging confirmed the EJV consistently traverses anterior to the clavicle near the sternoclavicular joint in mice, representing the first anatomically conserved landmark for this

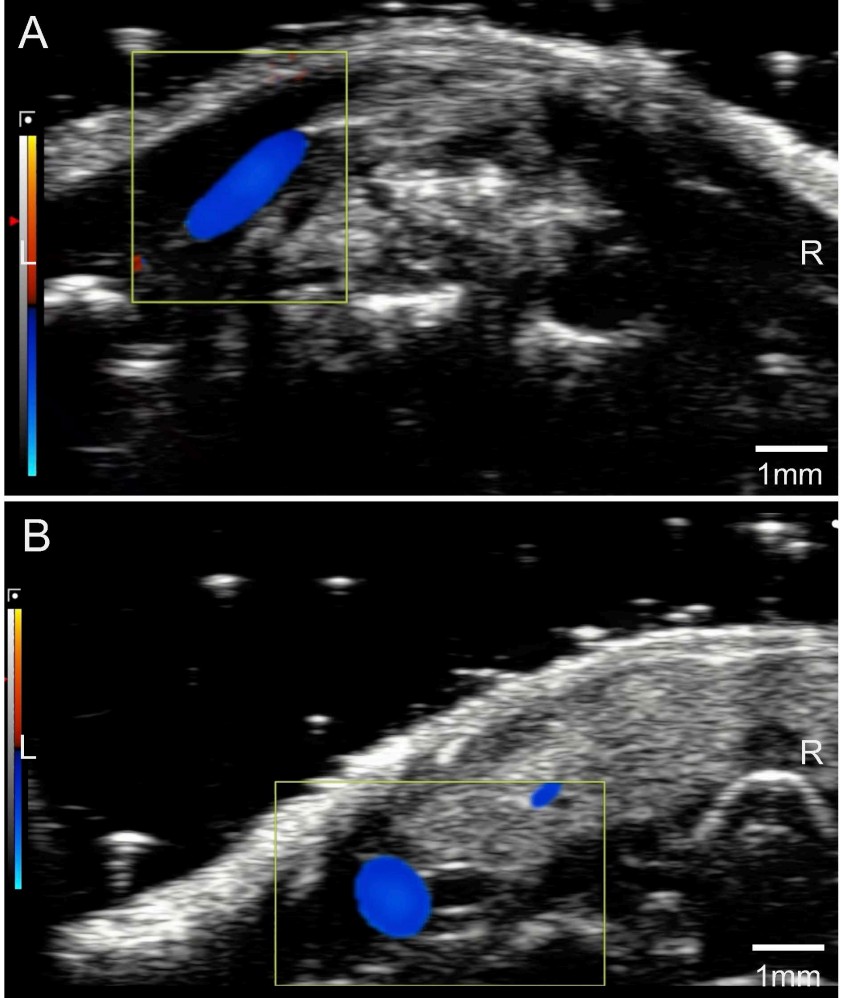

**Fig 1. Evaluation of the external jugular vein with high-resolution ultrasound imaging.** (A) Transverse view of the external jugular vein. (B) Longitudinal view of the external jugular vein. The external jugular vein traverses the superficial cervical fascia, lateral to the sternocleidomastoid muscle. This anatomical position, devoid of adjacent major vascular structures, was identified as the optimal site for venipuncture to minimize the risk of iatrogenic injury to surrounding tissues.

procedure (Fig 4). Comparative anatomical analysis shows a fundamental difference between mice and humans in the EJV course, which has significant implications for the development of accurate murine models in vascular research. Our detailed mapping of the EJV trajectory substantially advances our current understanding of murine cervicothoracic venous anatomy, with significant implications for experimental procedures requiring vascular access mice. The standardization of anatomical illustrations derived from this mapping will facilitate the widespread adoption and implementation of these methodological advancements. This study identified specific cutaneous landmarks that consistently predicted the optimal site for EJV cannulation, characterized by maximal vessel diameter and the most favorable trajectory for successful venous access. Central venous access in mice is often avoided due to concerns about severe complications associated with the procedure. The cervicothoracic region contains critical structures near central veins, requiring extreme precision during catheterization. Accidental puncture of major blood vessels, cardiac tissue, or pulmonary structures in this complex area can lead to rapid hemodynamic collapse and mortality [38]. High-precision 3D imaging revealed that the murine EJV

*PLOS One* | https://doi.org/10.1371/journal.pone.0329811    September 25, 2025                                                                 5 / 15

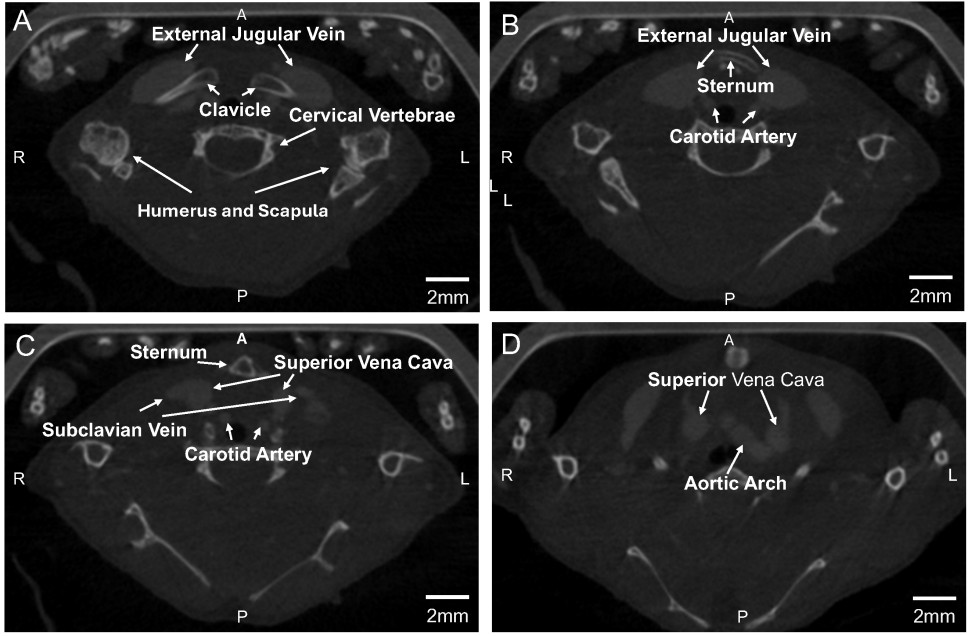

**Fig 2. Contrast CT of murine external jugular vein anatomy.** (A) The external jugular vein descends anteromedially across the clavicle, reaching its maximal diameter at this juncture. (B) The bilateral external jugular veins course from the cranial aspect of the sternum, traversing the sternoclavicular joints en route to the thoracic cavity. (C) The external jugular vein enters the thoracic cavity between the clavicle and first rib, subsequently joining the subclavian vein to form the brachiocephalic vein, which ultimately contributes to the superior vena cava. (D) The common carotid artery courses posterior to the external jugular vein, paralleling the trachea from its origin in the aortic arch.

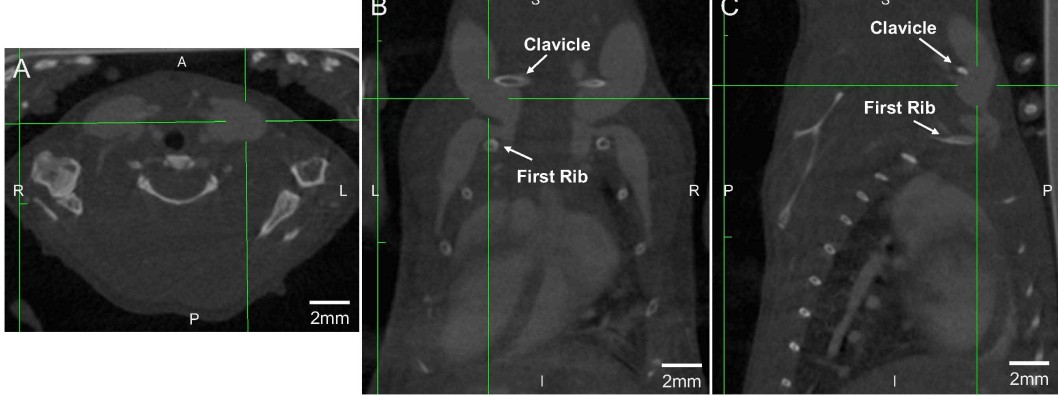

**Fig 3. Multi-planar reconstruction imaging of murine external jugular vein for optimized venipuncture.** (A) axial cross-section. (B) coronal cross-section. (C) sagittal cross-section. The intersection of the green lines in each section shows the left external jugular vein as the target of the venipuncture. The axial, coronal, and sagittal views of views demonstrate the superficial course of the external jugular vein as it traverses anteromedially across the clavicle before entering the thoracic cavity between the first ribs. The optimal needle trajectory follows the subclavian border, using the sternum, left clavicle, and sternoclavicular joint as anatomical landmarks.

travels superficially and anteriorly to the clavicle, a configuration distinct from humans (Fig 4). This anatomical insight allows for a venipuncture approach that avoids the carotid and subclavian arteries, thereby significantly reducing the risk of inadvertent arterial puncture. Following sternum and bilateral rib removal, comprehensive necropsy evaluation of the

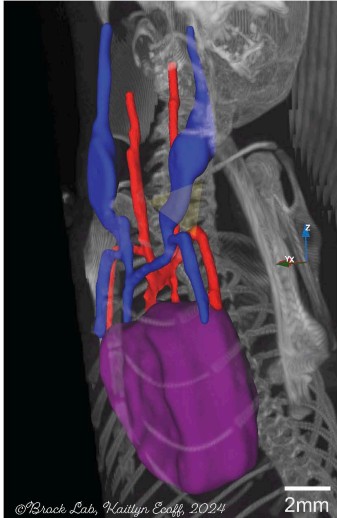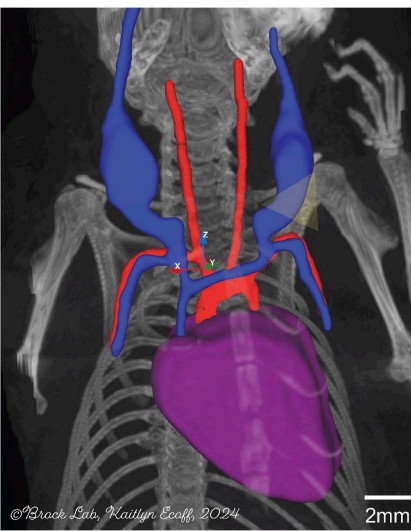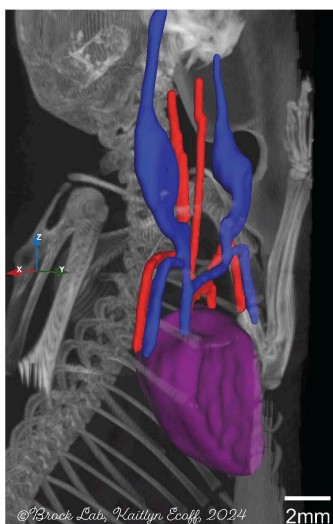

**Fig 4. 3D volume rendering of the external jugular vein and adjacent vasculature.** Contrast-enhanced CT data were used to generate a 3D volume rendering along the x, y, and z axes. This technique allows for detailed visualization of the external jugular vein and surrounding major blood vessels, providing a comprehensive view of the cervical vascular anatomy. Blue, red, and purple represent veins, arteries, and cardiac structures, respectively. The external jugular vein is shown at its maximal length superior to the clavicle. The subclavian vein, which is notably thinner than the external jugular vein and courses alongside the subclavian artery, is not recommended for venipuncture. A yellow triangle demarcates the surface projection of the external jugular vein, as inferred from external anatomical landmarks. The proposed puncture route follows the inferior border of the clavicle, allowing for reliable access to the external jugular vein while avoiding major cervicothoracic vasculature.

thoracic cavity clearly delineated the course of the EJV and its anatomical relationships with vulnerable structures such as the lungs and heart (Fig 5). Our anatomical analysis guided the development of a standardized venipuncture protocol, specifying precise needle trajectory, insertion angle, and depth. Adhering to these parameters minimizes the risk of fatal complications associated with inadvertent injury to vital structures.

To evaluate the procedural safety and efficacy of STEM, we conducted a multi-faceted validation process. The operator was not involved in the review procedure to ensure objectivity in the evaluation of procedural outcomes. The initial validation phase involved analyzing the learning curve and procedural outcomes during the novice period, as assessed by the technique developer (Table 1). A sample size calculation was performed to determine the number of mice needed to detect a significant reduction in the number of venipuncture attempts required for successful blood collection. Based on preliminary data (n = 5) showing an initial number of 3.0 ± 1.5 attempts, we aimed to detect a reduction to 1.5 attempts. Using a significance level of 0.05, a desired power of 80%, and an estimated effect size of 1.0, the calculation indicated that a minimum of 8 mice would be required. To account for potential data loss and increase the robustness of our findings, we chose to include 10 mice in the study. A total of 40 male and female matched C57BL/6J mice were used for an analysis of surgical outcomes and complications. Three cohorts of mice (n = 10 per cohort) were evaluated at different stages of protocol implementation: the initial cohort at protocol induction, a second cohort during early implementation, and a third cohort following the researchers' acquisition of extensive experience (>30 venipunctures performed). All procedures were done in accordance with the established protocol using the left EJV and with a targeted blood collection of 0.5% of the animal's body weight. The number of punctures decreased significantly from 3 [2-3] in the first cohort to 1 [1-1] after completing 30 venipunctures (p = 0.001, Dunn's test with Bonferroni correction). Similarly, the procedure time was substantially reduced from 81.0 ± 19.5 seconds in the first cohort to 56.2 ± 8.5 seconds after 30 venipunctures (p = 0.005, Tukey's HSD test). In all cohorts, a target blood collection volume of at least 100 µl was ensured (p = 0.489, ANOVA). Throughout the study period, including the initial learning phase, no mortality or severe adverse events were observed in

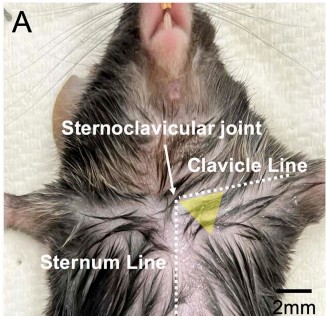
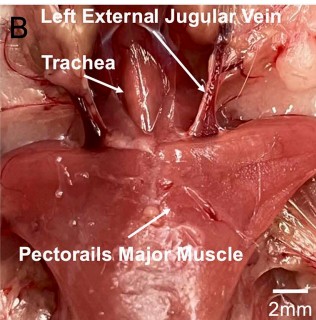
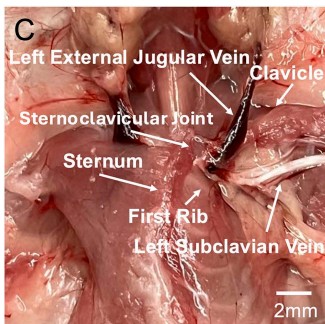
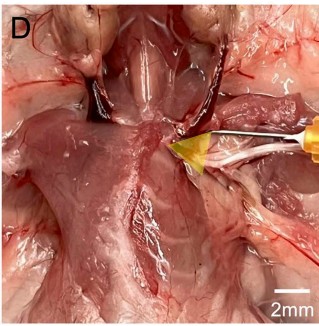
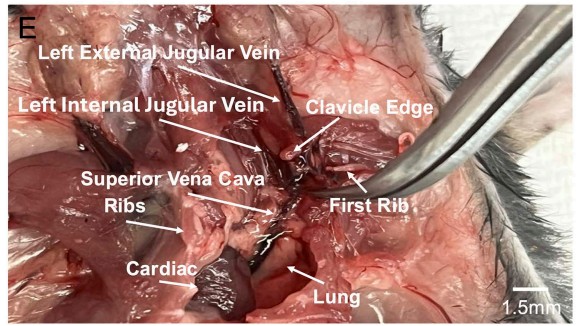

**Fig 5. Autopsy findings.** (A) C57BL/6J Mice (n = 5) were placed in the supine position. White dotted lines indicate the sternum and clavicle. Yellow triangles denote the proposed puncture sites for external jugular vein access. (B) Dissection with skin and submandibular glands removed, revealing bilateral external jugular veins. (C) Further dissection with left pectoralis major muscle removed, exposing the external jugular vein, clavicle, and subclavian vein. Note the oblique course of the external jugular vein as it crosses the clavicle and descends to enter the thoracic cavity near the first rib. (D) The anatomic findings of Fig 5D highlighting the optimal puncture site (yellow triangle) and preferred needle trajectory for external jugular vein access. (E) Thoracic cavity exposed after removal of the sternum and partial rib resection. The external jugular vein is shown joining the internal jugular vein before entering the superior vena cava, which returns blood to the heart.

any of the experimental animals. This technique was established in C57BL/6J mice but proved to be applicable to other strains of mice, including BALB/cJ, NOD/ShiLtJ, and DBA/2J (Fig 6, S3 Table).

The second validation phase evaluated the efficacy and safety of STEM for longitudinal blood collection (Table 2). Twenty-five female C57BL/6J mice underwent serial blood draws equivalent to 1% of body weight every 4 weeks, for a total of four collections to analyze procedural success rates and complications. The sample size was determined based on a learning curve study in which 30 different mice underwent a single procedure with a 100% success rate. To demonstrate the feasibility of achieving four successful longitudinal procedures in the same mouse, we assumed a conservative success rate of 99% with a 5% margin of error and a 95% confidence level. This calculation indicated that at least 16 mice would be required. To account for the stricter condition of repeated procedures, potential complication, and data loss, we increased the sample size to 25 mice. The left EJV was the primary target for venipuncture, with the right EJV serving as an alternative if left-sided access was unsuccessful. All mice (n = 25) successfully underwent four serial blood draws, each equivalent to 1% of body weight, at 4-week intervals. During the study period, one mouse required venipuncture of the right external jugular vein due to difficulty accessing the left side. A single instance of minor bleeding occurred, which was easily controlled with manual compression. No severe adverse events or mortality were observed throughout the longitudinal blood collection protocol. Repeated longitudinal blood sampling from the left EJV was feasible without altering the established landmarks or technique. STEM mitigates the peripheral vein damage typically associated with repeated sampling from conventional sites such as the lateral tail vein. In a series of over 300 venipunctures performed using STEM, no mortality was observed, and complications were limited to minor hemorrhages in <1% of procedures, demonstrating the technique's safety and reproducibility.

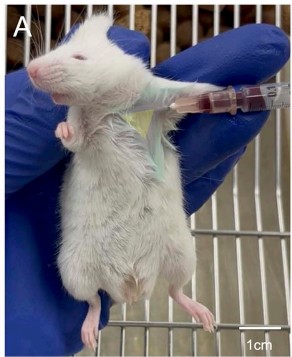
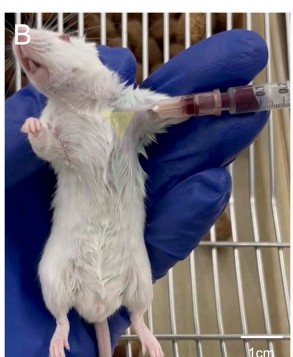
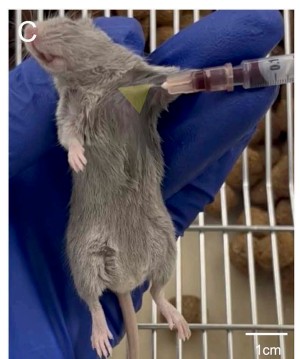
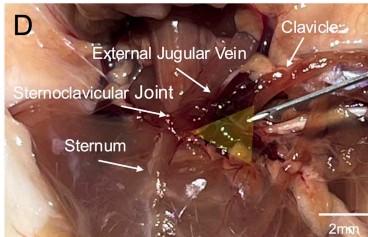
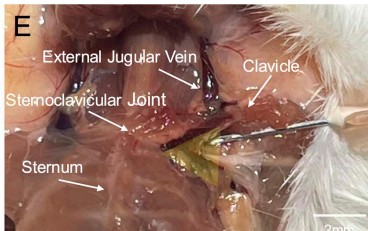
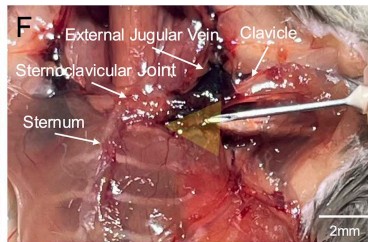

**Fig 6. Anatomical differences in the external jugular vein among mouse strains.** (A), (B) and (C) are images of actual STEM procedures in BALB/cJ, NOD/ShiLtJ, and DBA/2J, respectively. (D), (E) and (F) verify the anatomical similarity of EJV trajectories among different mouse strains based on autopsy findings. BALB/cJ, NOD/ShiLtJ, and DBA/2J also closely resemble C57BL/6J at the anatomy site where the EJV traverses the anterior clavicle in proximity to the sternoclavicular joint. The yellow triangle indicates the relationship between body surface landmarks and the optimal puncture site based on autopsy findings.

**Table 1. Procedural learning curve analysis for venipuncture: Outcomes during the novice phase.**

| | Initial | Second | Third | P value | Pairwise comparisons |
|---|---|---|---|---|---|
| Sex (M:F) | 5:5 | 5:5 | 5:5 | N.S. | |
| Body weight (g)* | 23.4±3.0 | 23.9±2.5 | 23.6±3.2 | 0.932 | |
| # of Punctures** | 3 [23] | 2 [12] | 1 [11] | 0.002 | Initial vs. Second: p=0.166, Second vs. Third: p=0.333, First vs. Third: p=0.001 |
| Procedure time (sec)* | 81±19.5 | 68±17.5 | 56.2±8.5 | 0.006 | Initial vs. Second: p=0.209, Second vs. Third: p=0.204, First vs. Third: p=0.005 |
| Volume of blood drawn targeted at 0.5% of body weight (µl)* | 105±20.6 | 106±9.6 | 114±21.7 | 0.489 | |
| Complication (%) | 0 | 0 | 0 | N.S. | |
| 30 days survival (%) | 100 | 100 | 100 | N.S. | |

* Values are given as the mean±standard deviation. ** Values are given as median [IQR]. Kruskal-Wallis test with post-hoc Dunn's test (Bonferroni-corrected) was used for number of punctures, while one-way ANOVA with Tukey's HSD post-hoc test was applied for procedure time. Statistical significance was set at p<0.05. N.S.: Not significant

The third validation phase was a comparative analysis between STEM and other blood collection methods to verify its benefits. Ten C57BL/6J mice were used for blood collection in each of STEM, Tail venipuncture (TV), and Retro-orbital bleeding (ROB) procedures. The target blood collection volume was set at 100 µl, and the number of punctures allowed was limited to no more than five. Outcome measures were success rate (100 µl of blood collected), number of punctures attempted, operative time from the first needle or hematocrit collection tube puncture to the end of the required blood collection or completion of five puncture procedures, and any complications. Table 3 shows the procedural outcomes.

**Table 2. Outcome of longitudinal blood sampling.**

|  | 0 week | 4 week | 8 week | 12 week |
|---|---|---|---|---|
| Mouse number | 25 | 25 | 25 | 25 |
| Body weight (g) | 24.6±3.3 | 25.3±3.1 | 25.8±2.0 | 26.7±2.5 |
| The success rate of blood collection at 1% of body weight (%) | 100 | 100 | 100 | 100 |
| Change of approach from left to right side (n) | 0 | 1 | 0 | 0 |
| Complications (n) | 0 | 0 | 1 (minor bleeding) | 0 |

Values are givens as the mean±standard deviation.

**Table 3. Comparison of procedural outcomes and stress levels between STEM and other venipuncture methods.**

|  | TV | ROB | STEM | P value | Pairwise comparisons |
|---|---|---|---|---|---|
| Sex (M:F) | 5:5 | 5:5 | 5:5 | N.S. | |
| Body weight (g)* | 22.2±3.5 | 22.3±3.4 | 22.4±3.4 | 0.990 | |
| Success rate (%) | 70 | 80 | 100 | 0.186 | |
| Attempted punctures** | 2 [2–5] | 1 [1–2] | 1 [1–1] | <0.001 | TV vs. ROB: p=0.009, TV vs. STEM: p=0.001, ROB vs. STEM: p=1.00 |
| Procedure time (sec)** | 196 [184-362] | 37 [25-46] | 17 [14–20] | <0.001 | TV vs. ROB: p=0.052, TV vs. STEM: p<0.001, ROB vs. STEM: p=0.047 |
| Complication (%) | 0 | 0 | 0 | N.S. | |

* Values are given as the mean±standard deviation. ** Values are given as median [IQR]. Kruskal-Wallis test with post-hoc Dunn's test (Bonferroni-corrected) used for number of attempted punctures and procedure times. Statistical significance was set at p<0.05. N.S.: Not significant. TV: Tail venipuncture, ROB: Retro-orbital bleeding, STEM: Sternoclavicular joint-Targeted External jugular venipuncture Method

Success rates were 70% for TV, 80% for ROB, and 100% for STEM with no significant differences in efficacy between groups. Nevertheless, ROB (p=0.009, Dunn's test with Bonferroni correction) and STEM (p=0.001, Dunn's test with Bonferroni correction) had significantly fewer attempted punctures than TV. STEM significantly shortened procedure time compared to TV (p<0.001, Dunn's test with Bonferroni correction) as well as ROB (p=0.047, Dunn's test with Bonferroni correction). No complications were observed in the third validation phase. Stress levels in mice between blood collection methods were compared by analyzing plasma corticosterone levels, a widely accepted physiological marker of acute stress obtained from blood samples in each of six mice (Fig 7). Plasma corticosterone levels were 182 ng/ml [119–234] in TV, 174 ng/ml [88–273] in ROB, and 129 ng/ml [71–180] in STEM, with a trend toward lower plasma levels in STEM that were not statistically significant (p=0.413, Kruskal-Wallis test). Factors, including fewer number of punctures to collect blood, shorter restraint times, and shorter procedure times, may suggest that STEM can reduce stress in mice. However, this finding may provide only a partial assessment of animal welfare. Future studies should incorporate behavioral, recovery, and long-term indicators to enable a more comprehensive evaluation of animal welfare, including potential stress on mice.

The fourth validation phase examined whether STEM can be easily learned and practiced by novice users (Table 4). Five members of our research group with no prior experience in mouse blood collection nor any involvement in the development of STEM to perform the procedure. Although the five researchers who worked in this study had varying levels of experience with mice, none had ever performed a venipuncture. All researchers received a standardized one-hour training session on the same day, which included viewing the protocol with technical video in this article and reviewing this manuscript. Following training, each researcher attempted blood collection from the left EJV. Given their novice status in mouse venipuncture, successful blood collection by these researchers was defined as the visible aspiration of blood into the syringe, regardless of the volume obtained. All five researchers successfully collected blood, with an average of 2.8±1.4

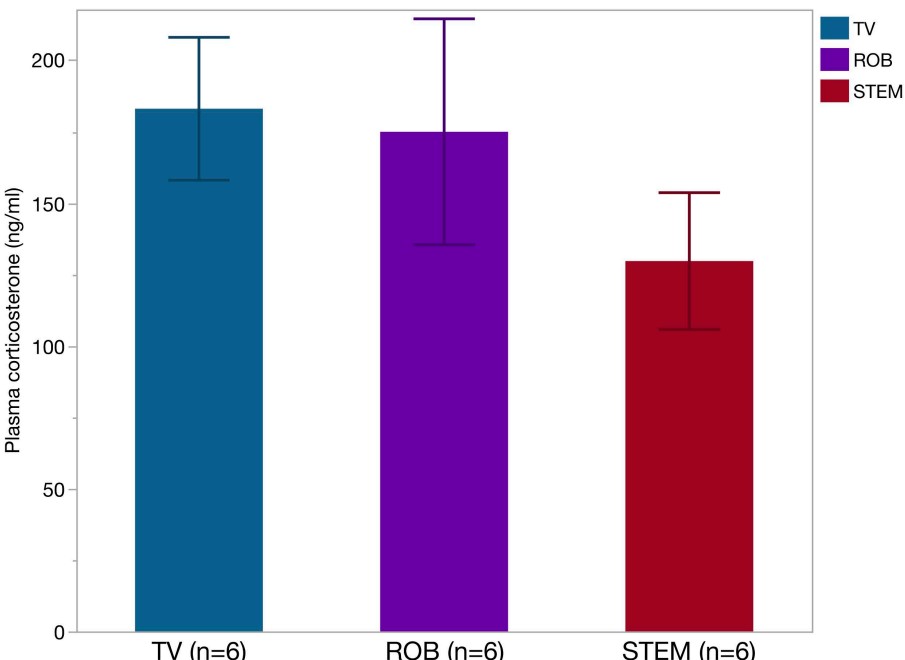

**Fig 7. Comparison of plasma corticosterone among blood collection methods.** Each bar graph is shown as median and SEM. Statistical analysis was performed using the Kruskal-Wallis test, and no significant differences were observed among the groups (p = 0.413). TV: Tail venipuncture, ROB: Retro-orbital bleeding, STEM: Sternoclavicular joint-Targeted External jugular venipuncture Method.

**Table 4. Skill acquisition for first-time learners.**

|  | n = 5 |
|---|---|
| Researchers background |  |
| Number of researchers with experience working with mice | 1 |
| Number of researchers with medical knowledge | 1 |
| Number of researchers with previous experience collecting blood in mice | 0 |
| Results of blood sampling |  |
| Puncture times | 2.8 ± 1.4 |
| Procedure time (sec) | 281 ± 149 |

Values are givens as the mean±standard deviation.

attempts required for success and a mean procedure time of 281 ± 149 seconds. One researcher with medical knowledge and an understanding of human anatomy successfully completed the procedure on the first attempt with a shorter execution time compared to other researchers. These findings indicate that novice operators can successfully obtain mouse blood samples by following the protocol detailed here, and that prior understanding of relevant anatomical structures may accelerate skill acquisition and procedural efficiency.

To compare the pharmacokinetics of IV administration via the EJV versus the lateral tail vein, indocyanine green (ICG, 0.5 mg/kg) was injected through each route respectively, in C57BL/6J mice (n = 5 per route), and ICG biodistribution was assessed using an IVIS Spectrum imaging system (Xenogen; Perkin Elmer, MA, USA) at 1 min, 30 min, 60 min, 180 min and 360 min post administration (Fig 8). Four Region of interests (ROIs) were defined: head, chest, abdomen, and tail.

In addition, ROIs were divided into puncture (chest and tail) and non-puncture regions (head and abdomen) for comparative analysis. The total radiant efficiency, a quantitative measure of fluorescence intensity specific to the IVIS system, was calculated for each ROI using Living Image software. This parameter allows for comparative analysis between routes of administration at each time point after ICG administration (Fig 8A). Rapid systemic distribution of ICG was observed in both groups, with detectable fluorescence in the head and peripheral extremities at 1-minute post-injection, indicating comparable biodistribution kinetics between EJV and lateral tail vein administration routes. Peak hepatic uptake of ICG occurred at approximately 30 minutes post-administration. ICG was then gradually excreted into bile and cleared through the intestines. The fluorescence signal in the head and extremities was undetectable after 360 minutes post-injection for both routes, indicating comparable systemic distribution and elimination. Total radiant efficiency at each time point after ICG administration in both puncture and non-puncture region was not significantly different between the EJV and lateral tail vein administration routes (Fig 8B, C) (P > 0.05 for all time points, Student t test). These results suggest that STEM

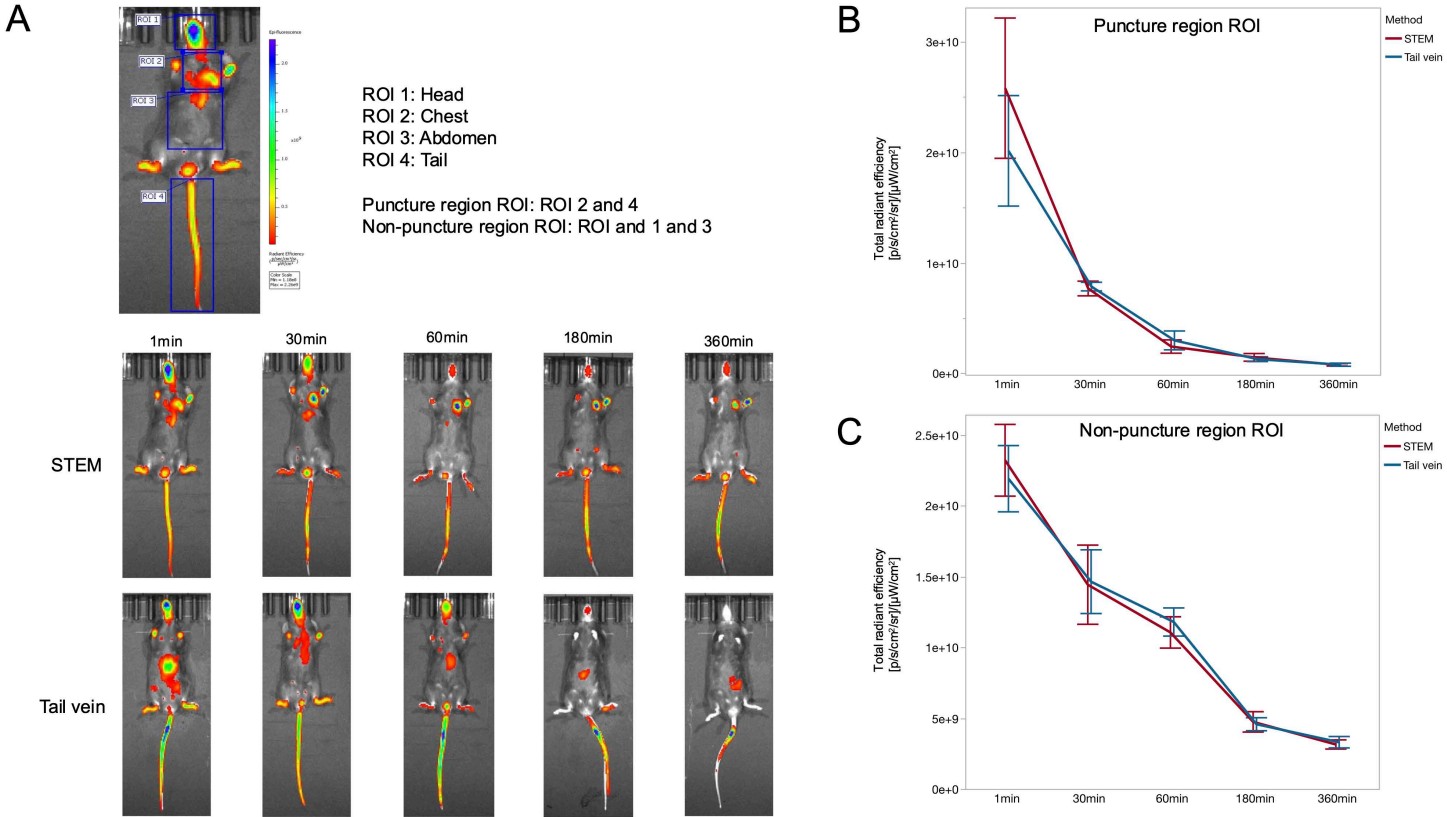

**Fig 8. Biodistribution and clearance kinetics of indocyanine green (ICG) following external jugular vein and lateral tail vein administration in mice.** Fluorescence imaging of ICG biodistribution in mice following intravenous administration via the external jugular vein or tail vein. (A) Total radiant efficiency for four ROIs was calculated and categorized into puncture and non-puncture regions for comparative analysis between routes of administration at each time point after ICG administration. Whole-body fluorescence was detectable within 1 minute post-injection with the signal visible in the limbs, tail, and head. This rapid and widespread distribution pattern provides strong evidence for efficient systemic circulation of the administered compound. At 30 minutes post-injection, the maximum hepatic accumulation of ICG was observed for both routes of administration. Time-course imaging revealed gradual hepatic metabolism and excretion of ICG, with near-complete clearance observed at 360 minutes post-injection, regardless of the initial administration route. (B) Total radiant efficiency at each time point after ICG administration in puncture region was not significantly different between administration routes (P > 0.05, Student t test). (C) Total radiant efficiency at each time point after ICG administration in non-puncture regions was not significantly different between administration routes (P > 0.05, Student t test). Error bars indicate standard error. ROI: Region of interest, STEM: Sternoclavicular joint-Targeted External jugular venipuncture Method.

did not alter the pharmacokinetics of intravenously administered drugs compared to tail vein administration, indicating no pharmacological advantage. Considering the procedural difficulties due to the size limitations of the tail vein, extravascular leakage of the drug due to the fragility of the tail vein vessel, and the feasibility of repeated administration, the value of STEM lies in its practical benefits, including ease of use, consistency of vascular access, and potential reduction in animal handling stress, rather than in modifications to drug kinetics.

There are several limitations to this study that should be considered. First, although our investigations were developed primarily in C57BL/6J mice, the anatomical similarities were confirmed in other mouse strains, BALB/c, NOD/ShiLtj, and DBA2J, and the applicability of the techniques was verified. However, strain-specific variations in vascular anatomy, vascular visibility, and response to venipuncture may affect the widespread adoption of this technique in other laboratory mouse strains. Although the success of this method with the well-known and commonly used C57BL/6J, BALB/c, NOD/ShiLtj, and DBA2J mice is noteworthy, further studies are needed to validate this approach across other mouse strains. In addition, our study was tested only in healthy adult mice, and demonstration of whether it works equally well in juvenile, aged, obese, and diseased animal models is also essential for expanding the applicability of STEM. Second, due to constraints such as animal welfare regulations, limited resources, and the exploratory nature of this study, our sample sizes were restricted. As a result, for the procedural comparison (success rate), corticosterone analysis, and pharmacokinetic studies, the number of animals may have been insufficient to reliably detect true differences between groups. Therefore, non-significant results in these analyses should be interpreted cautiously as evidence of equivalence, and further studies with larger cohorts will be needed to confirm these findings. Third, our results have not eliminated technical variability among operators, and the adaptability of this technique may need to take into account a learner's background of knowledge, skill, and experience. In addition, not all barriers to STEM acquisition for novices have been overcome. Additional enhancements of protocols and special training programs may be necessary during the implementation period to reduce the time and number of experiences required to master the technique. Another limitation is that all experiments were conducted in a single laboratory setting. Variations in equipment and institutional animal care practices may influence outcomes. Future multi-laboratory studies will be necessary to confirm reproducibility and generalizability of the STEM across different research environments.

In conclusion, we consider STEM to be a valuable and versatile venous access technique that has multiple advantages over other currently used blood collection methods. In time, we believe that STEM could become widely adopted for various mouse-based experiments in biomedical research.

## Supporting information

**S1 File. Sternoclavicular joint-Targeted External jugular venipuncture Method (STEM).** The step-by-step protocol is also available on protocol.io. https://dx.doi.org/10.17504/protocols.io.eq2lywqkevx9/v1.
(DOCX)

**S2 Text. Supplementary Methods.** The comprehensive imaging methodology implemented in this study and the experiments performed to establish the technique are described in detail.
(DOCX)

**S3 Table. Validation of STEM to mouse strains other than C57BL/6J.** Technical outcomes of STEM in BALB/cJ, NOD/ShiLtJ, and DBA/2J mice.
(DOCX)

## Acknowledgments

We thank S. Jain, K. Flavahan and O. Meze for their assistance with image capture and image editing. We thank R. Tsuchiya, A. Katagiri, S. Kenmochi, A. Omae and M. Abe for their support of our research.

## Author contributions

**Conceptualization:** Suguru Yamauchi, Malcolm V. Brock.

**Data curation:** John Michel, Jinny S. Ha, Takumi Iwasawa, Ryohma Tsuchiya, Miki Yamauchi.

**Funding acquisition:** Suguru Yamauchi, Andrei Gurau, Frank Bosmans, John Michel.

**Investigation:** Andrei Gurau, Kaitlyn Ecoff, Kristen P. Rodgers, Yuping Mei, Ryohma Tsuchiya, Miki Yamauchi.

**Methodology:** Suguru Yamauchi.

**Project administration:** Frank Bosmans, Franck Housseau, Yun Chen, Andreas S. Barth, Kazunori Kato, Hajime Orita, Shinji Mine, Tetsu Fukunaga, Malcolm V. Brock.

**Resources:** Kaitlyn Ecoff, Kristen P. Rodgers, Yuping Mei.

**Supervision:** Frank Bosmans, Franck Housseau, Malcolm V. Brock.

**Visualization:** Kaitlyn Ecoff, John Michel, Jinny S. Ha, Takumi Iwasawa.

**Writing – original draft:** Suguru Yamauchi, Malcolm V. Brock.

**Writing – review & editing:** Suguru Yamauchi, Frank Bosmans, Franck Housseau, Malcolm V. Brock.

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
