## [Decision Letter · Decision Letter 0]

29 Jan 2025

Dear Dr. Brock,

Thank you for submitting your manuscript to PLOS ONE. After careful consideration, we feel that it has merit but does not fully meet PLOS ONE’s publication criteria as it currently stands. Therefore, we invite you to submit a revised version of the manuscript that addresses the points raised during the review process.

We look forward to receiving your revised manuscript.

Kind regards,

Haipeng Liu

Academic Editor

PLOS ONE

Reviewers' comments:

Reviewer's Responses to Questions

**Comments to the Author**



Reviewer #1: Yes

Reviewer #2: Yes

2. Has the protocol been described in sufficient detail?

To answer this question, please click the link to protocols.io in the Materials and Methods section of the manuscript (if a link has been provided) or consult the step-by-step protocol in the Supporting Information files.

Reviewer #1: Yes

Reviewer #2: Partly

3. Does the protocol describe a validated method?

Reviewer #1: Yes

Reviewer #2: No

4. If the manuscript contains new data, have the authors made this data fully available?

Reviewer #1: N/A

Reviewer #2: Yes

**5. Is the article presented in an intelligible fashion and written in standard English?**

Reviewer #1: Yes

Reviewer #2: Yes

Reviewer #1: This paper introduces the Sternoclavicular joint-Targeted External Jugular Venipuncture Method (STEM), a novel technique for vascular access in mice. STEM leverages anatomical landmarks to provide reliable, reproducible venipuncture without requiring fur removal, deep sedation, or specialized restraints. It simplifies longitudinal blood sampling and intravenous injections, making the process more accessible to researchers of varying expertise. The study’s strengths lie in its detailed protocol, comprehensive imaging validation, focus on minimizing animal distress, and demonstration of novice-friendly learning.

While the method shows promise, further validation across diverse mouse strains and expanded statistical comparisons with other venipuncture techniques would strengthen its impact:

1. The study only used C57BL/6J mice. Expanding validation to other strains would enhance generalizability. Although the authors acknowledge this limitation, including preliminary data from additional strains could provide greater insights.

2. The manuscript reports means and standard deviations, but the learning curve data would benefit from inferential statistical tests (e.g., ANOVA) to determine the significance of differences between phases. Additionally, sections comparing EJV and tail vein injections could use more rigorous statistical analysis to confirm the lack of significant differences.

3. While the advantages over lateral tail vein injection are discussed, the manuscript could include direct comparisons of stress markers or physiological outcomes between methods. Evaluating the performance of STEM against other venous access points (e.g., retro-orbital bleeding) would further underscore its strengths.

Reviewer #2: The manuscript titled "A New External Jugular Venipuncture Technique for Efficient Vascular Access that Exploits a Murine Anatomical Variation" aims to present a novel method for central venous access in mice using a technique called STEM (Sternoclavicular joint-Targeted External jugular venipuncture Method). The study identifies a consistent anatomical feature in mice that enables reproducible and efficient venous access. While the manuscript addresses an important methodological challenge in preclinical research, there are substantial concerns that need to be addressed.

General Comments:

Strengths:

The manuscript highlights a novel technique for blood collection and intravenous access in mice, addressing a critical gap in preclinical methodology.

The use of a detailed anatomical analysis with advanced imaging provides strong visual support for the proposed approach.

High reproducibility and safety demonstrated in longitudinal studies underscore the potential utility of the method.

Weaknesses:

The lack of a robust comparative analysis with existing venous access methods limits the contextual significance of the findings.

The generalizability of the technique across other murine strains remains unvalidated.

There is minimal discussion of the translational relevance of this technique in broader preclinical research settings.

The manuscript’s claims about the ease of adoption for novice users are insufficiently supported by rigorous training outcome data.

Detailed Comments:

Introduction:

The introduction effectively outlines the challenges of venous access in mice but does not adequately justify why the STEM method is superior to current techniques such as lateral tail vein or retro-orbital venipuncture.

A more thorough literature review on existing venous access methodologies would provide a stronger rationale for this study.

Methods:

The study lacks detail on how anatomical variations across different mouse strains might impact the reproducibility of the technique. Validation in non-C57BL/6J strains is critical for generalization.

Provide more information on the sample size calculation and statistical power analysis to ensure that the study is sufficiently powered to draw conclusions.

The manuscript should explicitly state whether operators were blinded during the evaluation of success rates, procedural time, or complication rates.

Results:

The results emphasize the efficiency of STEM but do not compare this method against other established techniques in a head-to-head analysis. Including data on procedural success, time, and complications relative to existing methods would provide stronger evidence of STEM's advantages.

The training outcomes for novice users are underwhelming. The mean procedure time of 281 seconds with 2.8 attempts indicates that further optimization or training protocols may be required to ensure broader adoption.

The lack of significant differences in pharmacokinetics between the external jugular and lateral tail vein administrations raises questions about the added value of STEM beyond ease of use.

Discussion:

The discussion should include a more balanced assessment of the limitations of STEM, particularly its applicability to other mouse strains and potential operator variability.

Address the ethical implications of central venous access in mice, particularly concerning potential pain and stress despite claims of minimal sedation requirements.

While the study emphasizes the potential for STEM to replace traditional methods, there is insufficient evidence to support its widespread adoption without further comparative and validation studies.

Conclusion:

The conclusion should temper claims of STEM’s superiority and emphasize the need for further validation studies to establish its generalizability and translational relevance.

**Do you want your identity to be public for this peer review?** For information about this choice, including consent withdrawal, please see our Privacy Policy

Reviewer #1: No

Reviewer #2: No

---

## [Author Response · Author response to Decision Letter 1]

6 Mar 2025

Thank you for inviting us to submit a revised draft of our manuscript entitled, “A new external jugular venipuncture technique for efficient vascular access that exploits a murine anatomical variation” to PLOS ONE. We also appreciate the time and effort you and each of the reviewers have dedicated to providing insightful feedback on ways to strengthen our paper. Thus, it is with great pleasure that we resubmit our article for further consideration. We have incorporated changes that reflect the detailed suggestions you have provided. We also hope that our edits and the responses we provide below satisfactorily address all the issues and concerns you and the reviewers have noted.

To facilitate your review of our revisions, the following is a point-by-point response to the questions and comments. Revisions in the manuscript are indicated in red text.

Reviewer #1: Comment 1

The study only used C57BL/6J mice. Expanding validation to other strains would enhance generalizability. Although the authors acknowledge this limitation, including preliminary data from additional strains could provide greater insights.

Response: Thank you for providing these insights. We completely agree that this was a limitation, and so we now have extended our experiments to three additional strains of mice: BALB/cJ, NOD/ShiLtJ, and DBA2J. As a result, we are now able to demonstrate clearly the anatomical similarities of the external jugular vein among different mouse strains and the generalizability of STEM as a versatile technique. Please see the relevant section of the revised manuscript (Lines 216-218, 271-277, 336-337, 474-481; Fig 6 and S2 Table)

Reviewer #1: Comment 2

The manuscript reports means and standard deviations, but the learning curve data would benefit from inferential statistical tests (e.g., ANOVA) to determine the significance of differences between phases.

Response: Thank you for your suggestion. We have now performed the appropriate statistical analysis in the relevant sections (Lines 178-189, 329-334 and Table 1).

Reviewer #1: Comment 3

Additionally, sections comparing EJV and tail vein injections could use more rigorous statistical analysis to confirm the lack of significant differences.

Response: Thank you for your suggestion. We have incorporated your comments and have defined and analyzed the region of interest (ROI) by IVIS in order to clarify the differences in pharmacokinetics between the two IV administration methods. Please see the relevant changes in the revised manuscript (Lines 433-438, 445-453, 458-460 and 466-472.

Reviewer #1: Comment 4

While the advantages over lateral tail vein injection are discussed, the manuscript could include direct comparisons of stress markers or physiological outcomes between methods. Evaluating the performance of STEM against other venous access points (e.g., retro-orbital bleeding) would further underscore its strengths.

Response: Thank you for providing these valuable insights. We have incorporated your comments and made a direct comparison between surgical outcomes and plasma corticosterone measurements as a stress biomarker for STEM, Tail venipuncture (TV) and Retro-orbital bleeding (ROB). While we cannot completely rule out variance in the data due to the skill and experience of the operator, your suggestion of additional data does help to strengthen our argument. Please see the relevant edits in the revised manuscript (Lines 372-392, 394-400, 402-406, Fig 7 and Table 3)

Reviewer #2: Comment 1

The introduction effectively outlines the challenges of venous access in mice but does not adequately justify why the STEM method is superior to current techniques such as lateral tail vein or retro-orbital venipuncture. A more thorough literature review on existing venous access methodologies would provide a stronger rationale for this study.

Response: Thank you for your thoughtful review. We have added a literature review of current blood collection methods where applicable (Lines 90-117).

Reviewer #2: Comment 2

The study lacks detail on how anatomical variations across different mouse strains might impact the reproducibility of the technique. Validation in non-C57BL/6J strains is critical for generalization.

Response: Thank you for providing these insights. We completely agree and performed additional experiments on the mouse strains - BALB/cJ, NOD/ShiLtJ, and DBA2J. As a result, we were able to confirm the anatomical uniformity of the external jugular vein and the reproducibility of our STEM procedure among different mouse strains. As you pointed out, these new data provide strong evidence for the generalizability of our technique. Please see the relevant sections of the revised manuscript (Lines 216-218, 271-277, 336-337, 474-481; Fig 6 and S2 Table)

Reviewer #2: Comment 3

Provide more information on the sample size calculation and statistical power analysis to ensure that the study is sufficiently powered to draw conclusions. The manuscript should explicitly state whether operators were blinded during the evaluation of success rates, procedural time, or complication rates.

Response: Thank you for pointing out such an important deficiency in the manuscript. We have added a description in the relevant section (Lines 313-124, 316-322, and 349-355).

Reviewer #2: Comment 4

The results emphasize the efficiency of STEM but do not compare this method against other established techniques in a head-to-head analysis. Including data on procedural success, time, and complications relative to existing methods would provide stronger evidence of STEM's advantages.

Response: Thank you for providing these valuable insights. We performed a head-to-head analysis of STEM and other established methods, tail venipuncture (TV) and retro-orbital bleeding (ROB), with respect to procedural outcomes. Thanks to this suggestion, we now have evidence for the procedural superiority of STEM. Please see the relevant edits in the revised manuscript (Lines 372-392, 394-400, Fig 7 and Table 3)

Reviewer #2: Comment 5

The training outcomes for novice users are underwhelming. The mean procedure time of 281 seconds with 2.8 attempts indicates that further optimization or training protocols may be required to ensure broader adoption.

Response: Thank you for pointing this out. Your perspective is very important as we ponder how to show the general appeal of STEM to other researchers. We realize that we may not have addressed all the procedural learning and mastery issues at this time. We have added to the Discussion, comments describing the current limitations of training novices in the study and as a subject for future research (Lines 485-490).

Reviewer #2: Comment 6

The lack of significant differences in pharmacokinetics between the external jugular and lateral tail vein administrations raises questions about the added value of STEM beyond ease of use.

Response: Thanks for pointing that out. We don’t consider STEM's lack of pharmacokinetic differences in IV administration from the tail vein necessarily to be negative. Rather, the fact that STEM has demonstrated pharmacodynamics similar to IV administration from other established methods such as tail vein injection suggests that STEM can be applied to IV administration as well as blood collection. Moreover, STEM may be viewed as an alternative route of administration to a tail vein especially given its numerous benefits. We have added a description of this perspective, which we did not fully explain, in the relevant section (Lines 445-453 and Fig 8).

Reviewer #2: Comment 7

The discussion should include a more balanced assessment of the limitations of STEM, particularly its applicability to other mouse strains and potential operator variability.

Response: We agree with your point. We have been able to add results using other mouse strains and have updated the discussion accordingly. We have also included a reference to the variability in technique depending on the surgeon (Lines 474-481 and 485-490).

Reviewer #2: Comment 8

Address the ethical implications of central venous access in mice, particularly concerning potential pain and stress despite claims of minimal sedation requirements.

Response: Thank you for your suggestion. A comparative analysis of plasma corticosterone measurements as a stress marker among multiple blood collection methods, including STEM, is now included and demonstrates the validity of central venous access. We have added that description in the relevant section (Lines 387-392, Fig 7).

Reviewer #2's Comment 9

While the study emphasizes the potential for STEM to replace traditional methods, there is insufficient evidence to support its widespread adoption without further comparative and validation studies.

Response: Thank you for providing these valuable insights. We performed an experiment comparing STEM as a traditional blood collection method with tail venipuncture and retro-orbital bleeding and added the results of surgical outcomes and stress levels addressed in Comment 9. Although these results do not completely eliminate bias due to the surgeon's skill level and experience, we believe that they may provide some evidence for the feasibility of STEM on a broader scale. Please see the relevant section of the revised manuscript (Lines 372-392, 485-490, Fig 7 and Table 3)

Reviewer #2's Comment 10

The conclusion should temper claims of STEM’s superiority and emphasize the need for further validation studies to establish its generalizability and translational relevance.

Response: Thank you for your insightful comments. We were able to revise the conclusion of the manuscript with more appropriate phrasing (Lines 491-494).

We hope the revised version is now suitable for publication and look forward to hearing from you in due course.

---

## [Decision Letter · Decision Letter 1]

13 Aug 2025

Dear Dr. Brock,

We look forward to receiving your revised manuscript.

Kind regards,

Sarah Jose, Ph.D.

Staff Editor

PLOS ONE

Journal Requirements:

Reviewers' comments:

Reviewer's Responses to Questions

**Comments to the Author**



Reviewer #3: Yes

2. Has the protocol been described in sufficient detail?

To answer this question, please click the link to protocols.io in the Materials and Methods section of the manuscript (if a link has been provided) or consult the step-by-step protocol in the Supporting Information files.

Reviewer #3: Partly

3. Does the protocol describe a validated method?

Reviewer #3: Yes

4. If the manuscript contains new data, have the authors made this data fully available?

Reviewer #3: Yes

**5. Is the article presented in an intelligible fashion and written in standard English?**

Reviewer #3: Yes

Reviewer #3: The manuscript presents a protocol with clear utility to the research community, addressing a common procedural challenge in mouse studies and filling a gap in the published literature. The method is described in sufficient detail for an experienced researcher to reproduce it, with clear step-by-step guidance, figures, and methodological descriptions. Validation is supported by original data on procedural success rates, pharmacokinetic equivalence to tail vein injection, and stress assessment via corticosterone, though the scope is limited to healthy adult mice in one laboratory. All underlying data are made fully available in the supporting information, consistent with the PLOS Data policy. The manuscript is written in clear, standard English and is easy to follow, with only minor opportunities for improving readability by breaking up long sentences in the Discussion and briefly clarifying less common anatomical terms.

Below are specific key points which can further enhance the manuscript’s credibility.

1. The paper mentions a small sample size but does not explain that some key analyses may not have had enough statistical power to detect real differences. This includes the pharmacokinetic test with five animals per group, the corticosterone test with six animals per method, and the procedural comparison with ten animals per method. While post hoc group comparisons were done, there was no power calculation to confirm these sample sizes could support claims of no difference. Without this clarification, readers might take non-significant results as proof the methods are equivalent. I understand these numbers may reflect practical constraints such as animal welfare regulations, resource limits, or the exploratory nature of the study. Ideally, the authors could add a short post hoc power estimate for these results. If that is not possible, they should at least state clearly in the discussion that these specific analyses were underpowered and that a lack of significance should be interpreted with caution.

2. The study does not mention that all the experiments were carried out in a single laboratory. There is no discussion of whether the method would perform the same way in other facilities with different operators, equipment, or handling practices. Techniques that depend on identifying anatomical landmarks can be sensitive to user experience and handling style. Without inter-laboratory testing, it is unclear how reproducible the method would be elsewhere. I understand that expanding the work to multiple sites can be challenging due to coordination, time, and resource limits, especially in early-stage studies. Ideally, the authors could encourage future cross-lab testing to confirm reproducibility. If this cannot be added, at least noting in the discussion that this was a single-institution study would help set appropriate expectations for readers.

3. The study relies only on corticosterone levels to assess stress. While this is a standard and practical acute stress biomarker, it does not capture behavioral, recovery, or long-term welfare aspects. A single physiological measure can miss other impacts on the animals, and a more complete welfare profile would strengthen the method’s ethical claims. I understand that adding behavioral or long-term measures would require extra time, staff, and possibly new ethical approvals, which may not have been feasible in this study. Ideally, the authors could note this limitation and suggest that future work includes additional welfare indicators. At the very least, the discussion could explain why corticosterone was chosen as the sole measure and state that it provides only a partial picture of welfare.

4. The method was tested only in healthy adult mice from four strains. There is no discussion of whether it would work as well in juvenile, aged, obese, or diseased animals, where anatomy or vein accessibility may differ. Applicability to a wider range of ages and health conditions is important for a method proposed for broad use. Although the study did not state this age group as an objective, using healthy adults is a common choice for proof-of-concept work because they are physiologically stable and widely used in research. Still, it would help to note clearly that these results may not apply to other life stages or disease models, and to suggest that future studies test the method under those conditions. This would better define the scope and limits of the current findings.

5. The pharmacokinetic results showed no significant difference between STEM and tail vein injection. In the discussion, this is framed as a positive finding, suggesting STEM is equally suitable for IV drug administration. However, it is not acknowledged that this also means STEM offers no pharmacokinetic advantage over the comparator. Interpreting a lack of difference only as a benefit could give an incomplete picture of the findings. Though showing equivalence is still valuable, it would help to note explicitly that STEM does not alter drug kinetics compared to tail vein injection, so its advantages are likely practical (ease, consistency, reduced handling stress) rather than pharmacological. This clarification would make the conclusions more balanced and transparent.

**Do you want your identity to be public for this peer review?** For information about this choice, including consent withdrawal, please see our Privacy Policy

Reviewer #3: **Yes: ** Hector Espiritu

---

## [Author Response · Author response to Decision Letter 2]

1 Sep 2025

Response to Reviewers

August 22, 2025

Dear Editors,

Thank you for inviting us to submit a revised draft of our manuscript entitled, “A new external jugular venipuncture technique for efficient vascular access that exploits a murine anatomical variation” to PLOS ONE. We also appreciate the time and effort you and each of the reviewers have dedicated to providing insightful feedback on ways to strengthen our paper. Thus, it is with great pleasure that we resubmit our article for further consideration. We have incorporated changes that reflect the detailed suggestions you have provided. We also hope that our edits and the responses we provide below satisfactorily address all the issues and concerns you and the reviewers have noted.

Reviewer #3: The manuscript presents a protocol with clear utility to the research community, addressing a common procedural challenge in mouse studies and filling a gap in the published literature. The method is described in sufficient detail for an experienced researcher to reproduce it, with clear step-by-step guidance, figures, and methodological descriptions. Validation is supported by original data on procedural success rates, pharmacokinetic equivalence to tail vein injection, and stress assessment via corticosterone, though the scope is limited to healthy adult mice in one laboratory. All underlying data are made fully available in the supporting information, consistent with the PLOS Data policy. The manuscript is written in clear, standard English and is easy to follow, with only minor opportunities for improving readability by breaking up long sentences in the Discussion and briefly clarifying less common anatomical terms.

Below are specific key points which can further enhance the manuscript’s credibility.

1. The paper mentions a small sample size but does not explain that some key analyses may not have had enough statistical power to detect real differences. This includes the pharmacokinetic test with five animals per group, the corticosterone test with six animals per method, and the procedural comparison with ten animals per method. While post hoc group comparisons were done, there was no power calculation to confirm these sample sizes could support claims of no difference. Without this clarification, readers might take non-significant results as proof the methods are equivalent. I understand these numbers may reflect practical constraints such as animal welfare regulations, resource limits, or the exploratory nature of the study. Ideally, the authors could add a short post hoc power estimate for these results. If that is not possible, they should at least state clearly in the discussion that these specific analyses were underpowered and that a lack of significance should be interpreted with caution.

Response:

We thank the reviewer for this constructive feedback. We fully agree with the reviewer’s concern. Due to ethical and logistical considerations in animal use, our group sizes were limited, and we acknowledge that several comparisons (e.g., pharmacokinetics, corticosterone, and success rate of procedural outcomes) may have been underpowered. We have revised the Discussion to explicitly state that these results should be interpreted cautiously and that non-significant differences do not conclusively demonstrate equivalence (Lines 488-494).

2. The study does not mention that all the experiments were carried out in a single laboratory. There is no discussion of whether the method would perform the same way in other facilities with different operators, equipment, or handling practices. Techniques that depend on identifying anatomical landmarks can be sensitive to user experience and handling style. Without inter-laboratory testing, it is unclear how reproducible the method would be elsewhere. I understand that expanding the work to multiple sites can be challenging due to coordination, time, and resource limits, especially in early-stage studies. Ideally, the authors could encourage future cross-lab testing to confirm reproducibility. If this cannot be added, at least noting in the discussion that this was a single-institution study would help set appropriate expectations for readers.

Response:

We thank the reviewer for raising this important point. As you pointed out, assuring the reproducibility of a lab-independent technique is important to demonstrate the general adaptability of STEM. We have revised the Discussion to emphasize this limitation and to note that inter-laboratory validation will be essential in the future (Lines 499-503).

3. The study relies only on corticosterone levels to assess stress. While this is a standard and practical acute stress biomarker, it does not capture behavioral, recovery, or long-term welfare aspects. A single physiological measure can miss other impacts on the animals, and a more complete welfare profile would strengthen the method’s ethical claims. I understand that adding behavioral or long-term measures would require extra time, staff, and possibly new ethical approvals, which may not have been feasible in this study. Ideally, the authors could note this limitation and suggest that future work includes additional welfare indicators. At the very least, the discussion could explain why corticosterone was chosen as the sole measure and state that it provides only a partial picture of welfare.

Response:

We appreciate this suggestion. As noted by the reviewer, corticosterone was selected as a practical biomarker of acute stress, but it does not capture behavioral or long-term effects. We have clarified in the Discussion that corticosterone alone gives only a partial picture and suggested that future studies integrate behavioral and long-term indicators of animal welfare (Line 387, Lines 392-395).

4. The method was tested only in healthy adult mice from four strains. There is no discussion of whether it would work as well in juvenile, aged, obese, or diseased animals, where anatomy or vein accessibility may differ. Applicability to a wider range of ages and health conditions is important for a method proposed for broad use. Although the study did not state this age group as an objective, using healthy adults is a common choice for proof-of-concept work because they are physiologically stable and widely used in research. Still, it would help to note clearly that these results may not apply to other life stages or disease models, and to suggest that future studies test the method under those conditions. This would better define the scope and limits of the current findings.

Response:

We agree with this limitation and have added text to clarify the restricted scope of our study and the need for validation in other physiological contexts (Lines 485-488).

5. The pharmacokinetic results showed no significant difference between STEM and tail vein injection. In the discussion, this is framed as a positive finding, suggesting STEM is equally suitable for IV drug administration. However, it is not acknowledged that this also means STEM offers no pharmacokinetic advantage over the comparator. Interpreting a lack of difference only as a benefit could give an incomplete picture of the findings. Though showing equivalence is still valuable, it would help to note explicitly that STEM does not alter drug kinetics compared to tail vein injection, so its advantages are likely practical (ease, consistency, reduced handling stress) rather than pharmacological. This clarification would make the conclusions more balanced and transparent.

Response:

We appreciate this insightful point. We have revised the Discussion to clarify that STEM confers no pharmacokinetic advantage over tail vein injection, and that its benefits are primarily practical—such as ease of use, consistency, and potentially reduced handling stress (Lines 451-453, Lines 455-457). Thanks to you, the conclusion could be stated as a more transparent, balanced, and neutral interpretation.

We hope the revised version is now suitable for publication and look forward to hearing from you in due course.

Sincerely,

Malcolm V. Brock, M.D.

The E.F. Gordon Professor of Surgery and Oncology

---

## [Editor Report · Decision Letter 2]

8 Sep 2025

A new external jugular venipuncture technique for efficient vascular access that exploits a murine anatomical variation

PONE-D-24-37665R2

Dear Dr. Brock,

We’re pleased to inform you that your manuscript has been judged scientifically suitable for publication and will be formally accepted for publication once it meets all outstanding technical requirements.

Kind regards,

Erik Su

Academic Editor

PLOS ONE
---

## [Editor Report · Acceptance letter]

PONE-D-24-37665R2

PLOS ONE

Dear Dr. Brock,

I'm pleased to inform you that your manuscript has been deemed suitable for publication in PLOS ONE. Congratulations! Your manuscript is now being handed over to our production team.

Kind regards,

on behalf of

Dr. Erik Su

Academic Editor

PLOS ONE